# Electrochemical Milling of Deep-Narrow Grooves on GH4169 Alloy Using Tube Electrode with Wedged End Face

**DOI:** 10.3390/mi13071051

**Published:** 2022-06-30

**Authors:** Zhisen Ye, Guilin Qiu, Xiaolei Chen

**Affiliations:** 1Guangzhou Key Laboratory of Nontraditional Machining and Equipment, Guangdong University of Technology, Guangzhou 510006, China; yezhisen_gdut@163.com (Z.Y.); ql2268554299@163.com (G.Q.); 2State Key Laboratory of Precision Electronic Manufacturing Technology and Equipment, Guangdong University of Technology, Guangzhou 510006, China

**Keywords:** deep-narrow groove, GH4169 alloy, electrochemical milling, wedged end face

## Abstract

Deep-narrow grooves (DNGs) of nickel-based alloy GH4169 are extensively used in aerospace industry. Electrochemical milling (EC-milling) can manufacture special structures including DNGs by controlling the moving path of simple tool, showing a flexible process with the advantages of high machining efficiency, regardless of material hardness, no residual stresses, burrs, and tool wear. However, due to the inefficient removal of electrolytic by-products in the inter-electrode gap (IEG), the machining accuracy and surface quality are always unsatisfactory. In this paper, a novel tube tool with wedged end face is designed to generate pulsating flow field in IEG, which can enhance the removal of electrolytic by-products as well as improve the machining quality of DNG. The flow field simulation results show that the electrolyte velocity in the IEG is changed periodically along with the rotation of the tube tool. The pulsating amplitude of electrolyte is changed by adjusting the wedged angle in the end face of the tube tool, which could affect the EC-milling process. Experimental results suggest that the machining quality of DNG, including the average width, taper of sidewall, and surface roughness, is significantly improved by using the tube tool with wedged end face. Compared with other wedged angles, the end face with the wedged angle of 40° is more suitable for the EC-milling process. DNG with the width of 1.49 mm ± 0.04 mm, taper of 1.53° ± 0.46°, and surface roughness (Ra) of 1.04 μm is well manufactured with the milling rate of 0.42 mm/min. Moreover, increasing the spindle speed and feed rate can further improve the machining quality of DNG. Finally, a complex DNG structure with the depth of 5 mm is well manufactured with the spindle speed of 4000 rpm and feed rate of 0.48 mm/min.

## 1. Introduction

Due to their excellent fatigue resistance, corrosion resistance, radiation resistance, and high-temperature strength, Ni-based superalloys are used widely in die-casting molds, medical devices, and the aerospace sector as demand in the manufacturing field grows [1,2]. However, due to their material property of high-strength and poor heat conductivity, conventional machining often suffers from high tool wear, poor machining stability, and low process efficiency [3]. Nowadays, nontraditional processes with a low cost and high efficiency for manufacturing difficult-to-cut material are attracting more attention, including electrical discharge machining and laser machining. However, both cause heat affected layer and microcrack in machining surfaces, leading to lower machining quality [4,5].

Compared to other machining methods, electrochemical machining (ECM) has significant advantages for machining difficult-to-cut material, such as titanium alloys and Ni-based superalloys, due to the absence of tool wear, thermal and residual stresses, and cracks and burrs [6,7]. Electrochemical milling (EC-milling) combines the capabilities of ECM with the flexibility of numerical control (NC) technology, which can machine special structures, including deep-narrow grooves (DNGs), by controlling the moving path of the simple tool [8]. Natsu et al. fabricated three-dimensional complicated structures in SUS304 stainless using a tube electrode with an inner diameter of 0.26 mm by superimposing simple patterns [9]. Mitchell-Smith et al. produced different groove geometries by optimizing the bottom structure of a metallic nozzle to change current density distribution [10]. Liu et al. proposed a method of ultrasonic vibration-assisted electrochemical milling, which was helpful to improve the machined surface quality, and a three-dimensional structure with the total depth of 300 μm was machined [11]. Liu and Qu analyzed the anodic polarization curves of TB6 in NaNO_3_ solution and the effect of current density on the morphology, and grooves and flat surfaces were machined successfully by electrochemical milling [12]. Zhu et al. established the electrochemical milling with nanosecond pulse model and machined 2D and 3D complex structures with good shape precision and surface quality [13]. Rathod et al. used sidewall insulation electrode to prevent the dissolution of the material along the sidewall of microgroove and reduced the taper angle from 58.39° to 25.20° for microgroove [14].

In the above studies, the electrochemical milling process is started from an initial gap between the bottom of the electrode and the surface of the workpiece, in which the electric field is mainly provided by the bottom of the electrode. With a motion along the feed direction, a very thin layer of material is removed from the surface of the workpiece. As a result, when a high-aspect-ratio structure is machined with this method, the tube tool is needed to multipass feed along the depth and length directions. Due to the stray corrosion, electric fields are repeatedly formed at the machined zone during the multipass feed, the machining accuracy and surface quality are always unsatisfactory. In an attempt to achieve better machining profile, Ghoshal and Bhattacharyya proved that the taper angle of grooves generated by sinking and milling method was far less than layer-by-layer method and then proposed a reversed taper tube tool for electrochemical milling to reduce the taper angle of groove [15]. This clearly showed that one-pass milling with the tube tool is more suitable for machining DNG structure.

However, the electric field for electrochemical anodic dissolution is provided by the sidewall of tool electrode in this method, in which large amounts of electrolytic by-products (sludge, gas bubbles, and heat) are generated in the inter-electrode gap (IEG). When the machining gap becomes small, it is difficult to remove them from the IEG. To maintain the stability of electrolyte flow during the whole machining process, a rapid removal of the electrolytic by-products is needed in the deep and narrow IEG. If the removal capacity is insufficient, the machining accuracy and surface quality will become poor. To obtain high machining accuracy and machining stability, Wang et al. suggested electrochemical machining with vibration superimposed to improve the removal of sludge and hydrogen bubbles [16]. Bilgi et al. proposed an ECM process with rotating electrode movement to enhance the uniformity of electrolyte flow and reduce or eliminate the flow field disrupting processes [17]. Moreover, Niu et al. proposed a flow channel structure with six slits in sidewall for improving the uniformity of flow field, and a thin-wall structure with the depth of 3 mm was machined [18]. However, the machining accuracy and surface quality in high-aspect ratio structures by EC-milling still need to be improved.

Pulsating flow is one of the unsteady flows that can change the characteristics of hydrodynamics and enhance mass transport [19,20]. This paper proposes a method for using a tube tool with wedged end face to generate pulsating flow field in the IEG to enhance the removal of electrolytic by-products, as well as improve the machining accuracy and surface quality in EC-milling. The parameters of pulsating flow field could be changed by adjusting the wedged angle in the end face of the tube tool, which would affect the EC-milling process. The flow field distributions of different wedged angles in the end face of the tube tool are simulated numerically. Experiments also are conducted to investigate the effects of different wedged angles in the end face of the tube tool on the machining quality of DNG.

## 2. Description of the Method and Numerical Simulation

### 2.1. Description of the Method

Figure 1 shows a schematic diagram of EC-milling of DNG with a wedged end face tube electrode. The milling procedure is divided into two processes. First, the tube electrode with wedged end face is fed along the *Z*-axis to the required machining depth through electrochemical drilling (Figure 1a). Second, the workpiece then is controlled to move along the specified rail in the *X-Y* plane (Figure 1b). The electrolyte flows from the bottom of the tube electrode into the IEG, thereby dissolving the workpiece material while simultaneously flushing out the electrolytic by-products.

As shown in Figure 1b, the front sidewall of the tube electrode provides an electric field for electrochemical anodic dissolution, and the amounts of electrolytic by-products are generated in the IEG. By using the wedged end face of the tube tool in the experiment, the quantity and flow rate of electrolyte flowing into IEG will be changed periodically with the rotation of the tube tool, and the pulsating flow field is generated, which could enhance the removal of electrolytic by-products. In addition, the pulsating parameters of electrolyte, including pulsating amplitude and pulsating frequency, can be changed by adjusting the wedged angle (α) and rotating speed of the tube tool, which will affect the EC-milling process.

In the EC-milling process, the evolution process of DNG is illustrated in Figure 2. When the machining process is in a stable state, the front gap *Δx* in X direction can be described as [21]:(1)$$\Delta x=\frac{\eta \omega \kappa U_{R}}{v_{x}}$$
where *η* is the current efficiency, *ω* is the volumetric electrochemical equivalent of the metallic material, *κ* is the electric conductivity of electrolyte, *U_R_* is the supply voltage, and *v_x_* is the feed rate of the tube tool.

As shown in Figure 2a, when the tube tool is fed along the milling path in the plan *X-Y*, the relationship between the side gap and the *Y*-axis can be expressed as
(2)$$\frac{dy}{dt}=\frac{\eta \omega \kappa U_{R}}{y}$$

At the initial time, *t* = 0 and *y* = *y*_0_. Equation (2) can be integrated as follow:(3)$$\frac{y^{2}}{2}=\eta \omega \kappa U_{R}t+\frac{{y_{0}}^{2}}{2}$$

As the initial side gap, *y*_0_ can be assumed to be the same with the front gap *Δx*, the machining side gap *Δ**y* in the *Y* direction can be expressed as:(4)$$\Delta y=\sqrt{\eta \omega \kappa U_{R}t+\Delta x^{2}}=\sqrt{\eta \omega \kappa U_{R}\frac{D}{v_{x}}+\Delta x^{2}}$$
where *D* is the outer diameter of the tube tool.

Thus, the groove width *W* can be described as:(5)$$W=D+2\Delta y=D+2\sqrt{\eta \omega \kappa U_{R}\frac{D}{v_{x}}+\Delta x^{2}}$$

As shown in Equation (5), it can be obtained that the width of DNG is affected by numerous parameters, including the outer diameter of the tube tool, feed rate, supply voltage, and electrolyte conductivity. During machining, the workpiece is dissolved. Meanwhile, hydrogen and oxygen are generated on the cathode and anode surfaces, respectively. The electrolyte in the IEG will be warmed by Joule heat. All these factors interactively influence the distribution of electrolyte conductivity along the flow path, resulting in a taper sidewall, as illustrated in Figure 2b. The relationship among the electrolyte conductivity *κ*, electrolyte temperature *T*, and gas void fraction *β_gas_* can be described as follows:(6)$$\kappa =\kappa_{0}{\left(1-\beta_{gas}\right)}^{bp}\left(1+\alpha \left(T-T_{0}\right)\right)\;$$
where *κ*_0_ is the initial electrolyte conductivity, *T*_0_ is the initial electrolyte temperature, *α* is the degree of temperature dependence, and *bp* is Bruggeman’s coefficient.

Equation (6) shows that the efficient removal of electrolytic by-products (sludge, gas bubbles, and heat) is helpful to reduce the difference of electrolyte conductivity along the depth direction of DNG; thus, the taper of DNG’s sidewall can be reduced.

### 2.2. Numerical Simulation

In this research, different wedged angles in the end face of the tube tool were used for generating different pulsating parameters for the electrolyte, and the flow field distribution in IEG was analyzed by computational fluid dynamics (CFD).

#### 2.2.1. Model Building

Three-dimensional simulation model based on the flow field is established, as shown in Figure 3. The fluid domain includes the inner region of the tube electrode and DNG. Due to the rotation of the tube electrode, it needs to divide the fluid domain into a stationary zone and a rotation zone. The pink area represents the stationary zone, and the green area represents the rotation zone, which are transmitted data through the interface. In addition, the walls of the tube electrode are set as moving walls. Since the designed 3D model is relatively complex, tetrahedral meshes are used for the simulation model, and the mesh refinement is performed on the rotation zone and the around zone to ensure the accuracy of simulation results. In order to simplify the above model, this paper makes the following assumptions:(1)Electrolyte is a continuous incompressible viscous fluid.(2)The energy dissipation caused by the change of medium temperature and temperature difference is ignored in the machining process, and the flow is constrained by the conservation equation of mass and momentum.(3)The flow field is not affected by bubbles or particles.

For incompressible viscous fluids, the fluid flow in the turbulent state is restricted by the Navier–Stokes equation:(7)$$\nabla \cdot \bar{v}=0$$
(8)$$\frac{\partial \bar{v}}{\partial t}+\left(\bar{v}\cdot \nabla \right)\bar{v}=-\frac{1}{\rho }\nabla p+\mu \nabla^{2}\bar{v}$$
where *ρ* is the electrolyte density, *v* is the flow velocity, *p* is the pressure, and *μ* is the dynamic viscosity of the electrolyte.

Based on the change of the flow field, the *k-ε* turbulence model in the standard equation is used to solve the turbulent energy *k* and the turbulent dissipation rate *ε* in the electrolyte flow process:(9)$$\frac{\partial \left(\rho k\right)}{\partial t}+\frac{\partial \left(\rho ku_{i}\right)}{\partial x_{i}}=\frac{\partial }{\partial x_{j}}\left[\left(\mu +\frac{\mu_{t}}{\sigma_{k}}\right)\frac{\partial k}{\partial x_{j}}\right]+G_{k}-\rho \varepsilon$$
(10)$$\frac{\partial \left(\rho \varepsilon \right)}{\partial t}+\frac{\partial \left(\rho \varepsilon u_{i}\right)}{\partial x_{i}}=\frac{\partial }{\partial x_{j}}\left[\left(\mu +\frac{\mu_{t}}{\sigma_{\varepsilon }}\right)\frac{\partial k}{\partial x_{j}}\right]+\frac{C_{1\varepsilon }}{k}G_{k}-C_{2\varepsilon }\rho \frac{\varepsilon^{2}}{k}$$
where *P_k_* is the generating term of turbulent energy, *σ_k_* and *σ_ε_* are Prandtl numbers corresponding to *k* and *ε* with values of 1.0 and 1.3, and *C_1ε_* and *C_2ε_* are model constants with values of 1.44 and 1.92.

All the models are solved by ANSYS FLUENT 19.2, and the simulation parameters are listed in Table 1.

#### 2.2.2. Simulation Results

Figure 4 shows the contour of flow velocity on the cross section with different wedged angles under different rotational degrees. In the standard flat end face model (see Figure 4a), the velocity of electrolyte in the IEG is constant. In contrast, it can be found that the velocity of electrolyte in both front IEG and side IEG is changed in the wedged end face models under different rotation degrees (Figure 4b–d). The simulation results indicate that the pulsating flow field can be well generated with the wedged end face in tube electrode. Meanwhile, with the increase in the wedged angle, the velocity change of electrolyte in the IEG is more obvious. In the wedged angle of 40° model, the maximum and minimum velocities of electrolyte in the IEG are about 30 m/s and 8 m/s. In the wedged angle of 60° model, the maximum and minimum velocities of electrolyte in the IEG are about 32 m/s and 0 m/s.

In order to further analyze the change of electrolyte flow rate in the IEG, points A to C marked in Figure 4a are referenced to describe the specific electrolyte velocity in the IEG. Figure 5 shows the change of flow velocity at points A to C under different tube tools. In the standard flat end face model, the value of flow velocity at points A to C is constant. The velocity of electrolyte in the front gap and side gap is 23.8 m/s and 15.3 m/s, respectively.

The trends of the velocity of pulsating field with different wedged angles are the same. There are two peaks of electrolyte velocity in one cycle in the front machining gap, while the velocity of electrolyte changes like a sine wave in the side machining gap. With the increase in the wedged angle, the pulsating amplitude of electrolyte increases. When the end face with the wedged angle is 40°, the velocity of electrolyte at point A ranges from 6.7 to 30.6 m/s. When the wedged angle increases to 60°, the velocity of electrolyte ranges from 0.3 to 32.8 m/s.

## 3. Experimental Section

Figure 6 shows a schematic of EC-milling system with a tube electrode. The experimental setup included the pulse power supply, electrolyte supply unit, and electrolysis devices. The workpiece was installed on an *X-Y* stage. The electrode tool was carefully attached at the terminal of the spindle, and then the spindle was installed on a *Z*-axis and driven by an inverter. The speed of the spindle was adjusted by controlling the output frequency of the inverter. In addition, the function of internal flushing was achieved by using a rotating joint, which could transfer electrolyte from a pipeline into a rotating spindle. Figure 7 shows a photograph of the tube electrodes with different end faces, which were made from stainless steel 304, and the outer diameter and inner diameter were 1.0 mm and 0.8 mm, respectively.

In this work, the workpiece material was GH4169, and the machining depth and length of the electrode were 5 mm and 10 mm, respectively. The machining accuracy of DNG was investigated by detecting the width and taper. The size measurement of DNG is shown in Figure 8, where five points along the length of a DNG were measured, the width was the sectional width on the top, and the taper was the angle between the vertical line and the sectional sidewall. The average values of the width and taper were obtained, and the SD (standard deviation) was used to evaluate the dimensional uniformity. The morphologies of DNGs were examined using a scanning electron microscope. The profiles and surface roughness (Ra) of DNGs were measured using a confocal laser-scanning microscope (CLSM, Olympus LEXT OLS4100, Tokyo, Japan) and a Step profiler (Kosaka, ET-150, Tokyo, Japan).

Experiments were performed on each proposed tool to analyze its effect for machining accuracy and surface quality in the electrochemical milling process. Moreover, the influence of other parameters on machining quality was explored by single-factor experiment. The machining parameters are listed in Table 2.

## 4. Results and Discussion

### 4.1. The Comparison of DNGs Generated with Different Wedged Angles

In order to compare the difference of DNGs generated with different wedged angles in the end face of the tube tool, comparative experiments were designed with the following machining parameters: the applied voltage of 25 V, pulse duty cycle of 50%, pulse frequency of 1 kHz, feed rate of 0.42 mm/min, and spindle speed of 3000 rpm.

Figure 9 and Figure 10 show the SEM images, dimensions, and 3D profiles of DNGs generated with different wedged angles in the end face of the tube tool. Using standard flat end face with the wedged angle of 0°, the average width and taper of the grooves are 1.57 mm ± 0.06 mm (mean ± SD) and 2.2° ± 0.43°, respectively. When using the end face with wedged angle of 40°, the average width and taper of DNG decrease to 1.49 mm ± 0.04 mm and 1.53° ± 0.46°. However, with the increase in the wedged angle, the average width and taper of DNG gradually increase, reaching to 1.60 mm ± 0.04 mm and 2.3° ± 0.45°, respectively, when the wedged angle reaches to 60°.

Figure 11 shows the comparison of the SEM images and surface roughness (Ra) of DNGs milled with different wedged angles. The surface roughness (Ra) of sidewall is 1.387 μm under the standard flat end face. When the wedged angle is 40°, the surface roughness (Ra) suddenly drops to 1.09 μm. When the wedged angle varies from 50° to 60°, the surface roughness (Ra) increases to 1.65 μm. The reason is that a pulsating electrolyte is generated in the IEG by using the wedged end face of the tube tool, which occurs as flow separation resulting in a large number of vortices at the wall surface and increasing the turbulence and mixing of fluid. The boundaries of this flow are characterized by the creation and destruction of eddies of large turbulence energy and vortex shedding, which helps the removal of electrolytic by-products and reduces the difference of electrolyte conductivity in the IEG along the depth direction of DNG [19]. Hence, the material dissolution rate becomes uniform, reducing the taper of DNG and improving the surface quality. In addition, due to the reduction in the accumulation of electrolytic by-products in the IEG, the flow resistance decreases, avoiding the accumulation of electrolyte at the edge of DNG and reducing the stray corrosion at the upper edge of DNG. Thus, the upper width of DNG decreases. With the increase in the wedged angle, the machining quality becomes poor. As the wedged angle increases, the electric field distribution on the sidewall becomes nonuniform, which will lead to the nonuniform electrochemical dissolution. Thus, the taper of DNG increases. Meanwhile, combined with flow field simulation results, the pulsating amplitude of electrolyte increases but the minimum velocity of electrolyte decreases to about 0 m/s with the increase in the wedged angle. There is a low velocity zone of electrolyte in the IEG. Thus, anodic dissolution occurs in an instant static flow, the electrolytic by-products cannot be removed from IEG, which is not beneficial to the machining process. In fact, many studies have reported that a pulsating flow with proper pulsating parameters is helpful to the transfer process [22].

### 4.2. The Effect of Spindle Speed on the Generation of DNGs

In this subsection, single-factor experiments were performed to investigate the effect of spindle speed of 2500, 3000, 3500, and 4000 rpm on the dimension by using tube electrode with wedged angle of 40°. Additionally, other machining parameters were set as the applied voltage of 25 V, feed rate of 0.42 mm/min, pulse frequency of 1 kHz, and pulse duty cycle of 50%.

Figure 12 and Figure 13 show the SEM images, dimensional change, and 3D profiles of DNGs generated with different spindle speeds. It can be observed that the average width of DNG decreases from 1.54 ± 0.04 mm to 1.49 ± 0.04 mm, and the average taper decreases from 1.62° ± 0.44° to 1.53° ± 0.46° when the spindle speed increases from 2500 rpm to 3000 rpm. As the spindle speed increases from 3000 to 4000 rpm, the average width maintains at about 1.5 mm, and the average taper gradually decreases from 1.53° ± 0.46° to 1.32° ± 0.32°. This result indicates that the spindle speed has no obvious influence on the width of DNG, while increasing spindle speed can improve the verticality of sidewall. High rotation is effective for the removal of electrolytic by-products from the IEG [23,24]. Meanwhile, the high pulsating frequency of electrolyte further improves the transfer of heat and electrolytic by-products, reducing the differences of electrolyte conductivity and material dissolution rate along the depth direction of DNG. Thus, the taper angle of DNG decreases. In addition, the spindle speed also affects the surface quality of DNGs. With the spindle speed increased, the pulsating frequency of electrolyte is increased, which could further enhance the removal of electrolytic by-products in IEG; thus, the electrochemical dissolution of material becomes more uniform, and the milled surface quality is improved. As shown in Figure 14, when the spindle speed increases from 2500 rpm to 4000 rpm, the surface roughness (Ra) of sidewall decreases from 1.14 μm to 0.92 μm.

### 4.3. The Effect of Feed Rate on the Generation of DNGs

In order to investigate the effect of the tube tool feed rate on the machining quality, single-factor experiments with the feed rate at 0.30, 0.36, 0.42, and 0.48 mm/min and the spindle speed at 4000 rpm were conducted. The other machining parameters were set as the applied voltage of 25 V, wedged angle of 40°, rotational speed of spindle of 4000 rpm, pulse frequency of 1 kHz, and pulse duty cycle of 50%.

Figure 15 and Figure 16 show the SEM images, dimensions, and 3D profiles of DNGs generated with different feed rates of tube electrode. As the feed rate increases from 0.30 to 0.48 mm/min, the average width of DNG decreases from 1.63 mm to 1.48 mm, and the average taper decreases from 2.66° to 1.32°. Moreover, the standard deviations of the width and taper of DNG decrease significantly with the increase in the feed rate. Thus, the consistency of the DNG enhances considerably.

According to Equation (1), with the increase in feed rate *v_x_* of the tube tool, the front gap *Δx* decreases. As shown in Equations (4) and (5), with the decrease in the front gap *Δx*, the width of groove *W* decreases. In addition, when the workpiece is moving toward the electrode, the electric field is provided mostly by the front of the electrode. However, the milled surface of the groove is inevitably exposed to the back of the tube electrode, and a small amount of material is dissolved from the milled surface of the DNG due to corrosion by the stray current. Thus, with the increase in feed rate, the time of secondary stray dissolution to sidewall of DNG reduces, and the taper of sidewall and its standard deviation gradually decreases. At the same time, the reduction in secondary stray dissolution to sidewall improves the surface quality of sidewall. As shown in Figure 17, when the feed rate increases from 0.30 to 0.48 mm/min, the surface roughness (Ra) decreases from 1.41 μm to 0.85 μm.

### 4.4. EC-Milling of Complex Narrow Grooves by Using Wedged Tube Electrode

Based on the above research, the complex deep-narrow groove structures with machining depth of 5 mm are machined on GH4169 nickel-based alloy in one-pass feed by using a wedged end face tube electrode with wedged angle of 40°, as shown in Figure 18. The machining parameters are a feed rate of 0.48 mm/min, applied voltage of 25 V, spindle speed of 4000 rpm, pulse frequency of 1 kHz, and pulse duty of 50%.

## 5. Conclusions

In this study, a novel tube electrode with wedged end face has been proposed for electrochemical milling, deep-narrow groove on GH4160 alloy and to improve the machining accuracy and surface quality. Based on simulations and experiments, the main conclusions can be obtained as follows:The simulation results indicated that the pulsating electrolyte could be generated in the inter-electrode gap by using a tube electrode with wedged end face. With the increase in wedged angle, the pulsating amplitude of electrolyte increased, but the minimum velocity electrolyte decreased.Experiments verified that a pulsating electrolyte generated by the wedged angle of 40° was more suitable for the EC-milling process, and both the machining accuracy and surface quality were improved. Additionally, the average width and taper of DNG was 1.49 mm ± 0.04 mm and 1.53° ± 0.46°, respectively. The surface roughness (Ra) of the sidewall reduced to 1.04 μm at the same time.The machining quality of DNG was improved by increasing the electrode rotational speed and feed rate. When the spindle speed was 4000 rpm and feed rate was 0.48 mm/min, the average width and taper of DNG was 1.48 mm and 1.32°, respectively. And the surface roughness (Ra) of sidewall was 0.85 μm.A complex deep-narrow groove structure with the depth of 5 mm was fabricated stably on a GH4169 nickel-based alloy in one-pass feed by using a wedged end face tube electrode with wedged angle of 40° at a spindle speed of 4000 rpm and feed rate of 0.48 mm/min.

## Figures and Tables

**Figure 1 micromachines-13-01051-f001:**
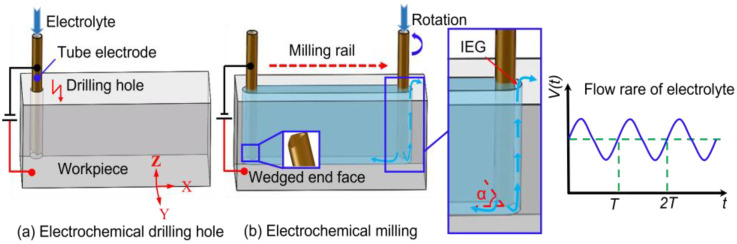
The schematic diagram of EC-milling of DNG with a wedged end face tube electrode. (**a**) Electrochemical drilling hole, (**b**) Electrochemical milling.

**Figure 2 micromachines-13-01051-f002:**
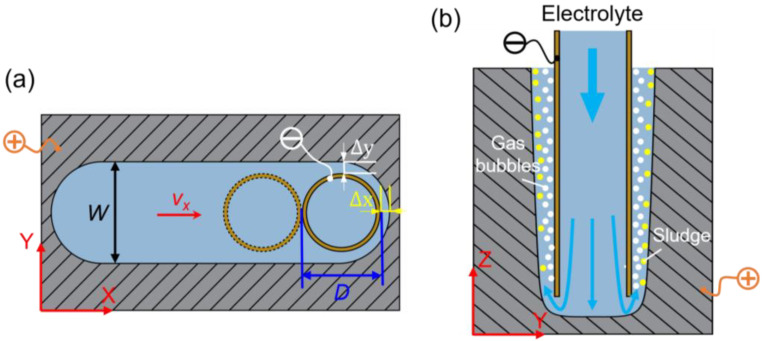
Diagram of the electrochemical milling process: (**a**) Sketch of the *X-Y* plane and (**b**) sketch of *Y-Z* plane.

**Figure 3 micromachines-13-01051-f003:**
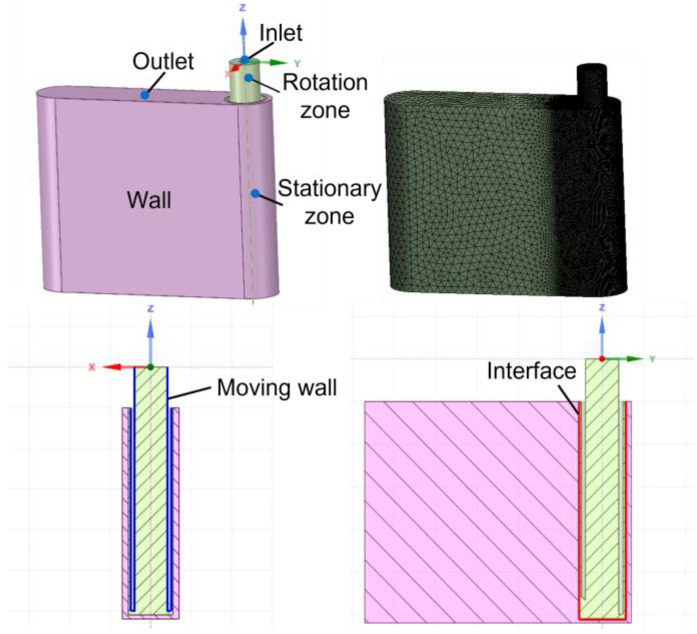
Building of three-dimensional model for the flow field simulation.

**Figure 4 micromachines-13-01051-f004:**
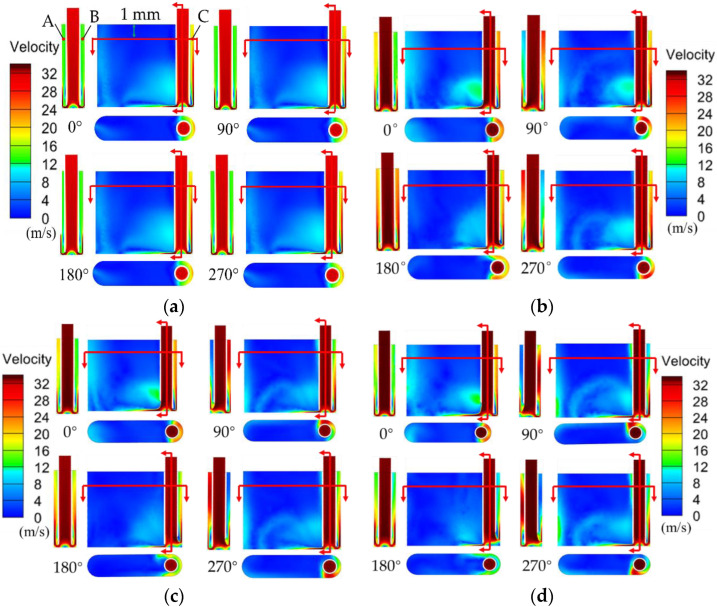
The contour of flow velocity on the cross section with different wedged angles in the end face of the tube tool. (**a**) 0°; (**b**) 40°; (**c**) 50°; (**d**) 60°.

**Figure 5 micromachines-13-01051-f005:**
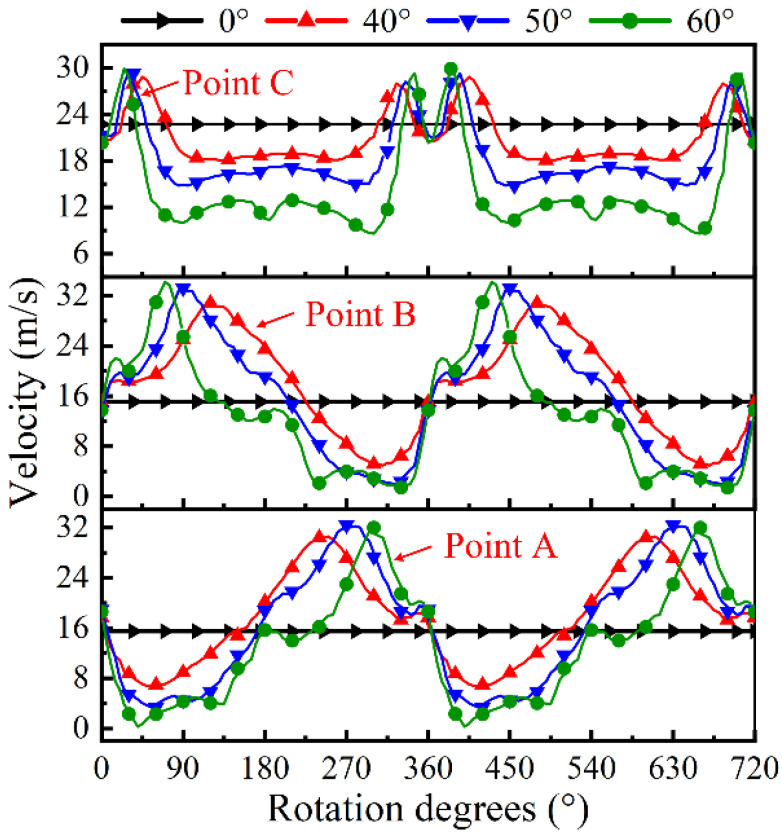
The variation of electrolyte velocity on points A to C in the IEG under kinds of tube tools.

**Figure 6 micromachines-13-01051-f006:**
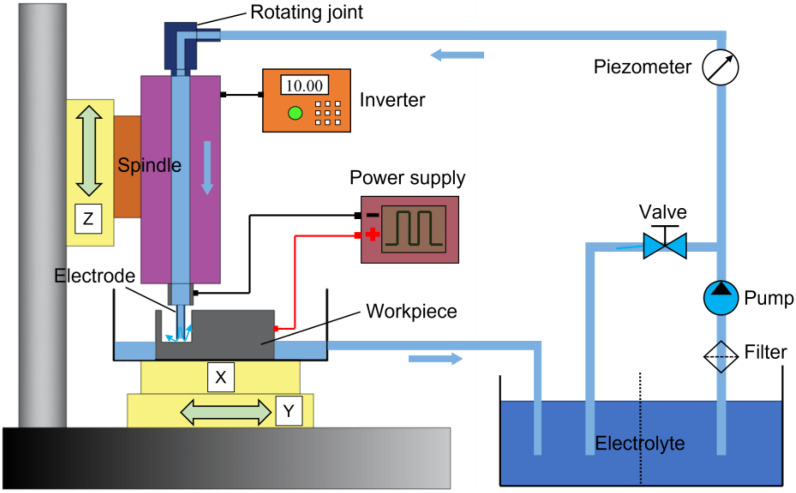
Schematic diagram of the experimental system.

**Figure 7 micromachines-13-01051-f007:**
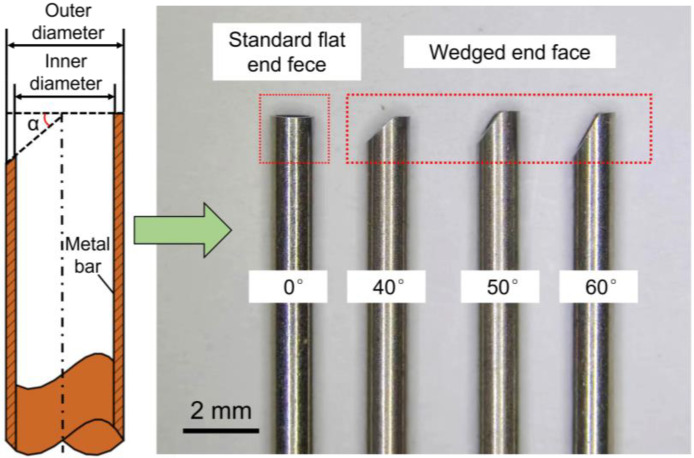
The photo of tube electrodes with different end faces.

**Figure 8 micromachines-13-01051-f008:**
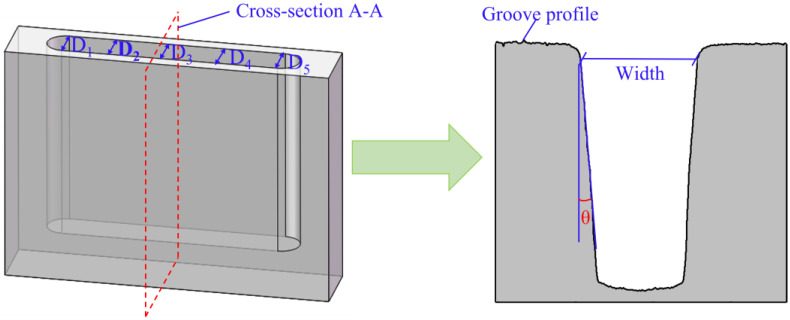
Schematic diagram of deep-narrow groove measurement.

**Figure 9 micromachines-13-01051-f009:**
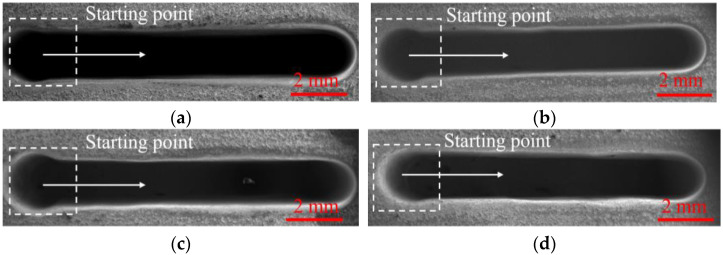
The SEM images of DNGs machined with different wedged angles in the end face of tube electrode. (**a**) Wedged angle = 0°; (**b**) wedged angle = 40°; (**c**) wedged angle = 50°; (**d**) wedged angle = 60°.

**Figure 10 micromachines-13-01051-f010:**
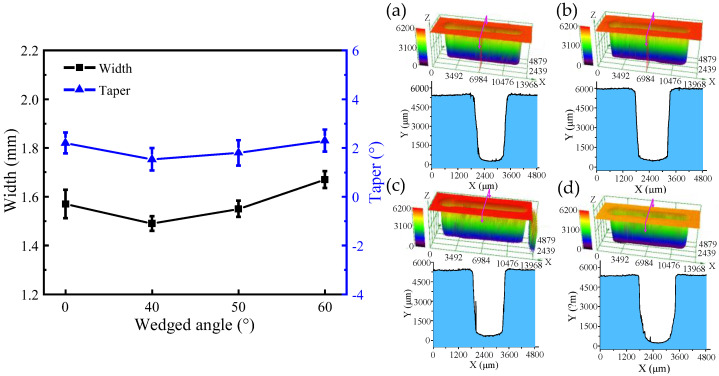
The dimension and 3D profile of DNGs milled with different wedged angles in the end face of the tube tool: (**a**) wedged angle = 0°; (**b**) wedged angle = 40°; (**c**) wedged angle = 50°; and (**d**) wedged angle = 60°.

**Figure 11 micromachines-13-01051-f011:**
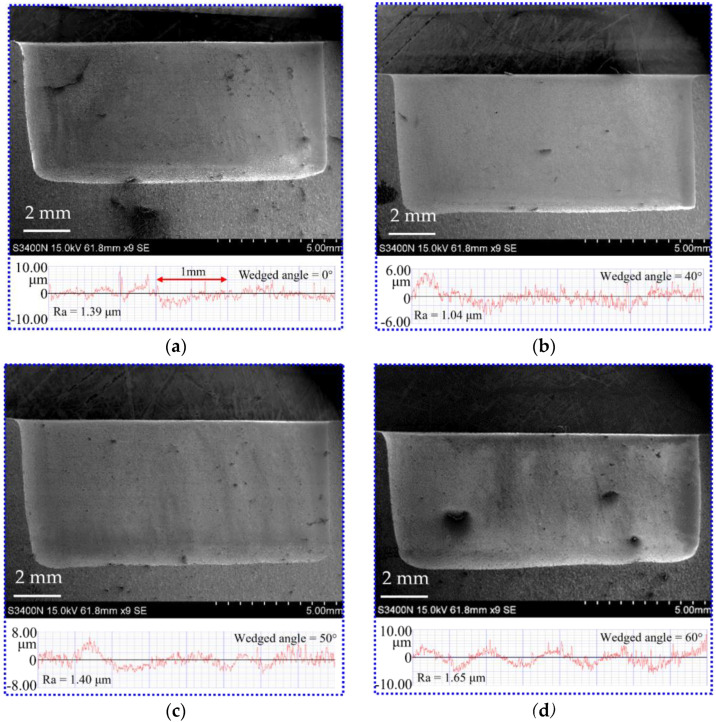
The SEM and surface roughness of sidewall of DNGs generated with different wedge angles in the end face of the tube tool: (**a**) wedged angle = 0°; (**b**) wedged angle = 40°; (**c**) wedged angle = 50°; and (**d**) wedged angle = 60°.

**Figure 12 micromachines-13-01051-f012:**
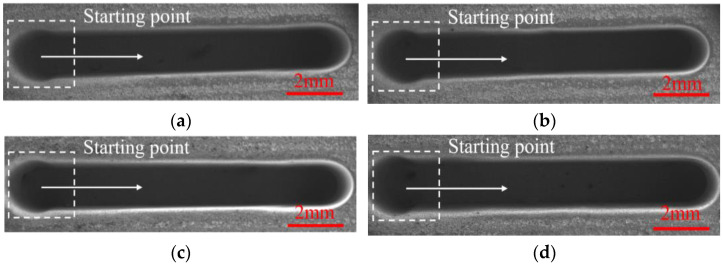
The SEM images of DNGs generated under different spindle speeds. (**a**) Spindle speed = 2500 rpm; (**b**) spindle speed =3000 rpm; (**c**) spindle speed = 3500 rpm; (**d**) spindle speed = 4000 rpm.

**Figure 13 micromachines-13-01051-f013:**
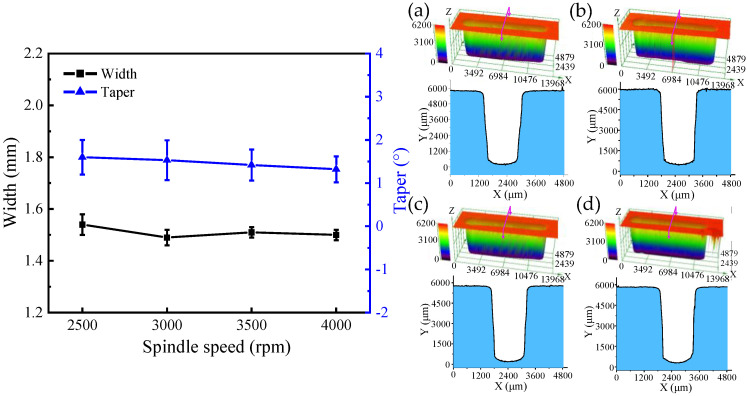
The dimension and 3D profile of DNGs milled with different spindle speeds: (**a**) spindle speed = 2500 rpm; (**b**) spindle speed = 3000 rpm; (**c**) spindle speed = 3500 rpm; and (**d**) spindle speed = 4000 rpm.

**Figure 14 micromachines-13-01051-f014:**
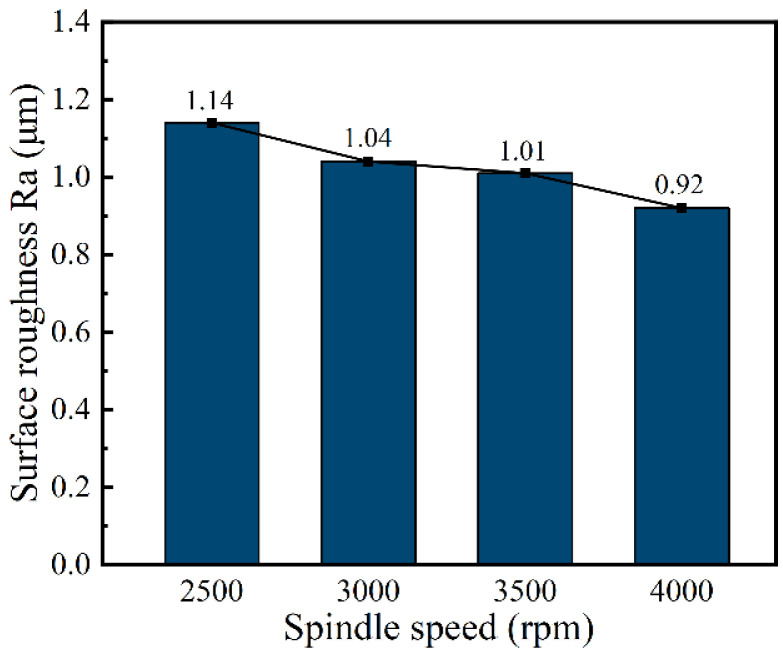
The surface roughness (Ra) of sidewall with different spindle speeds.

**Figure 15 micromachines-13-01051-f015:**
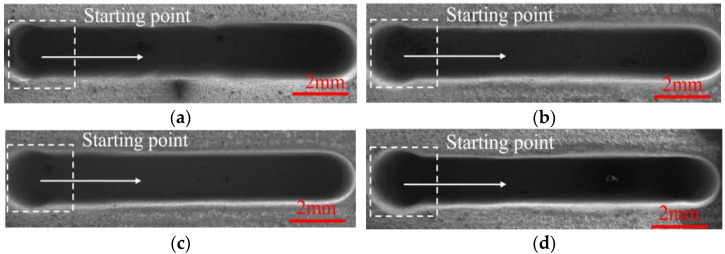
The SEM images of DNGs generated with varied feed rates. (**a**) Feed rate = 0.30 mm/min; (**b**) Feed rate = 0.36 mm/min; (**c**) Feed rate = 0.42 mm/min; (**d**) Feed rate = 0.48 mm/min.

**Figure 16 micromachines-13-01051-f016:**
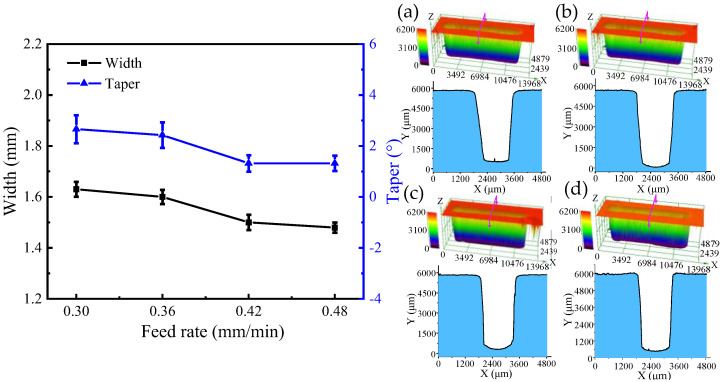
The dimension and 3D profile of DNGs milled with different feed rates: (**a**) feed rate = 0.30 mm/min; (**b**) feed rate = 0.36 mm/min; (**c**) feed rate = 0.42 mm/min; and (**d**) feed rate = 0.48 mm/min.

**Figure 17 micromachines-13-01051-f017:**
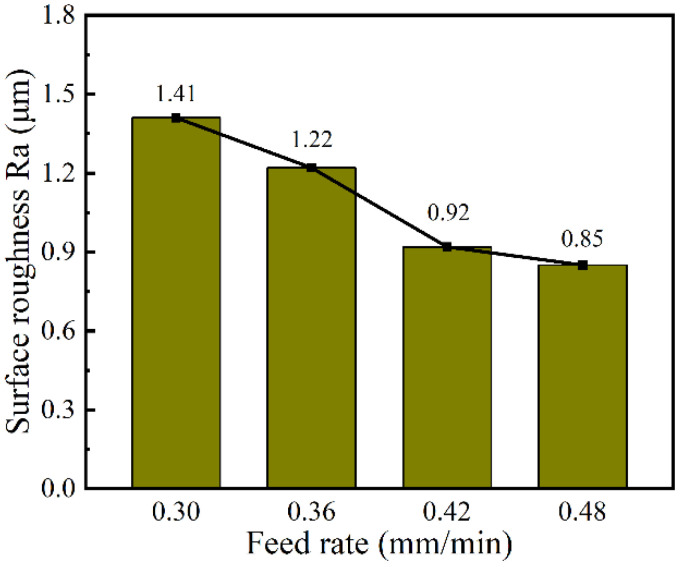
The surface roughness (Ra) of sidewall with varied feed rates.

**Figure 18 micromachines-13-01051-f018:**
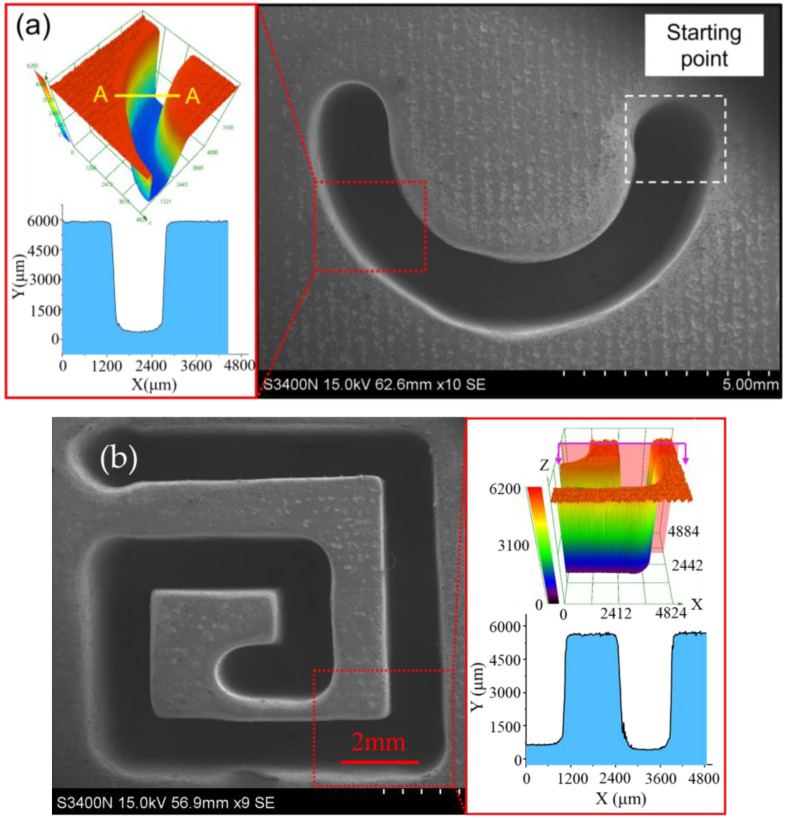
The SEM of complex DNG structures: (**a**) a semi-circular arc DNG structure and (**b**) a linear DNG structure.

**Table 1 micromachines-13-01051-t001:** The parameters for the simulation.

Parameter	Value
Inlet pressure, *p_in_*	1.2 MPa
Outlet pressure, *p_out_*	0 MPa
Rotational speed, *ω*	3000 rpm
Inter-electrode gap, *Δ*	0.25 mm
Dynamic viscosity of electrolyte, *μ*	1.003 × 10^−3^ Pa·s
Density of electrolyte, *ρ*	1100 kg/m^3^
External diameter of tube electrode, *D*	1 mm
Internal diameter of tube electrode, *d*	0.8 mm
Wedged angles, *α*	0°, 40°, 50°, 60°
Groove length, *L*	5 mm
Groove depth, *H*	5 mm

**Table 2 micromachines-13-01051-t002:** Machining parameters.

Parameters	Value
Electrolyte concentration	12% (wt.%), NaNO_3_
Electrolyte temperature	25 °C
Electrolyte pressure	1.2 MPa
External diameter of tube electrode	1 mm
Internal diameter of tube electrode	0.8 mm
Wedge angle	0°, 40°, 50°, 60°
Spindle speed	2500, 3000, 3500, 4000 rpm
Feed rate	0.30, 0.36, 0.42, 0.48 mm/min
Applied voltage	25 V
Pulse frequency	1 kHz
Pulse duty cycle	50%
Machining depth	5 mm
Machining length	10 mm
Material of workpiece	GH4169

## Data Availability

Not applicable.

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
