# Peer review of "Electrochemical Milling of Deep-Narrow Grooves on GH4169 Alloy Using Tube Electrode with Wedged End Face"

_micromachines, 2022, doi:10.3390/mi13071051_

Round 1

Reviewer 1 Report

A novel tube electrode with wedged end face has been proposed in this paper for  electrochemical milling deep-narrow groove on GH4160 alloy and to improve the machining accuracy and surface quality. I think this paper is innovative, the experimental data is detailed and reliable, and has a certain prospect of industrial application.  It is recommended  to accept and publish this paper in the micromachines journal.

Author Response

Thank you very much for your recognition of our work.

Reviewer 2 Report

Due to instead of Since– line 32

Dot missing – line 39

Providing instead of provide – line 75

2.1 Description of method. Please clarify if the tool with be rotating.

Phrase not clear - lines 115-116

Wrong English – lines 144-147

“More obvious” needs more explanation – line 189-190

The “reason” of the results needs theorical support – line 261-275

How is the surface finish affected? – line 296

Results do not verify the 40 degrees as there is no theoretical work or simulation showing this, please support your statement – line 350

Please explain: in the simulation 40 degrees tool shows less turbulence than larger values; why lower turbulence, i.e. less effective removal of heat and by-products, generates a better result?

Author Response

Question 1:

Due to instead of Since – line 32

Answer 1:

Thanks for your comment. We have revised the word:

Due to their excellent fatigue resistance, corrosion resistance, radiation resistance and high-temperature strength, Ni-based superalloys are widely in die-casting molds, medical devices and the aerospace sector as demand in the manufacturing field grows.

Question 2:

Dot missing – line 39

Answer 2:

Thanks for your comment. We have rechecked and added the missing.

Question 3:

Providing instead of provide – line 75

Answer 3:

Thanks for your comment. We have revised the sentences:

However, the electric field for electrochemical anodic dissolution is provided by the sidewall of electrode in this method, large amounts of electrolytic by-products (sludge, gas bubbles and heat) are generated in the inter-electrode gap (IEG).

Question 4:

2.1 Description of method. Please clarify if the tool with be rotating

Answer 4:

Thank you for your professional comment. To make it clear, we have rewritten the sentences:

By using the wedged end face tube tool in the experiment, the quantity and flow rate of electrolyte flowing into IEG will be changed periodically with the rotation of tube tool, and the pulsating flow field is generated, which could enhance the removal of electrolytic by-products.

Question 5:

Phrase not clear - lines 115-116

Answer 5:

Thanks for your comment. To make it clear, we have rewritten the sentences:

In addition, the pulsating parameters of electrolyte including pulsating amplitude and pulsating frequency can be changed by adjusting the wedged angle and rotating speed of tube tool, which will affect the EC-milling process.

Question 6:

Wrong English – lines 144-147

Answer 6:

Thanks for your professional comment. We have rewritten the sentences:

According to Equation (6), it can be observed that the efficient removal of electrolytic by-products (sludge, gas bubbles and heat) is helpful to reduce the difference of electrolyte conductivity along the depth direction of DNG, thus the taper of DNG’s sidewall can be reduced.

Question 7:

“More obvious” needs more explanation – line 189-190

Answer 7:

Thanks for your comment. To make it more understood, we have added the sentences:

Meanwhile, as the increase of wedged angle, the velocity change of electrolyte in the IEG is more obvious. In the wedged angle of 40° model, the maximum and minimum velocities of electrolyte in the IEG are about 30 m/s and 8 m/s. In the wedged angle of 60° model, the maximum and minimum velocities of electrolyte in the IEG are about 32m/s and 0 m/s.

Question 8:

The “reason” of the results needs theorical support – line 261-275

Answer 8:

Thanks for your professional comment. We have rewritten the sentences:

The reason can be explained that: a pulsating electrolyte is generated in the IEG by using the wedged end face tube tool, which occurs flow separation resulting in a larger number of vortices at the wall surface and increasing the turbulence and mixing of fluid. The boundaries of this flow are characterized by the creation and destruction of eddies of large turbulence energy and vortex shedding, which helps the removal of electrolytic by-products and reduces the difference of electrolyte conductivity in the IEG along the depth direction of DNG [19]. Hence, the material dissolution rate becomes uniform, reducing the taper of DNG and improving the surface quality. In addition, due to the reduction of the accumulation of electrolytic by-products in the IEG, the flow resistance decreases, avoiding the accumulation of electrolyte at the edge of DNG and reducing the stray corrosion at the upper of DNG. Thus, the upper width of DNG decreases. With the increase of wedged angle, the machining quality becomes poor. It can be explained that with the wedged angle increased, the electric field distribution on the side wall will become nun-uniform, which will lead to the nun-uniform electrochemical dissolution, thus the taper of DNG is increased. Meanwhile, combined with flow field simulation result, it can be indicated that the pulsating amplitude of electrolyte increases but the minimum velocity of electrolyte decreases even to about 0 m/s with the increase of wedged angle. There is a low velocity zone of electrolyte in the IEG. Thus, anodic dissolution occurred in an instant static flow, the electrolytic by-products cannot be removed out of IEG, which is not beneficial to the machining process. In fact, many studies have reported that a pulsating flow with proper pulsating parameters is helpful to the transfer process [22].

  1. Qu, N. S.; Fang, X. L.; Zhang, X. D.; Zhu, D. Enhancement of surface roughness in electrochamical machining of Ti6Al4V by pulsating electrolyte. Int. J. Adv. Manuf. Technol. 2013, 69, 2703-2709.

Question 9:

How is the surface finish affected? – line 296

Answer 9:

Thank you for your comment. To make it clear, we have rewritten the sentences:

In addition, the spindle speed also affects the surface quality of DNGs. With the spindle speed increased, the pulsating frequency of electrolyte is increased, which could further enhance the removal of electrolytic by-products in IEG, thus the electrochemical dissolution of material become more uniform, and the milled surface quality is improved. As shown in figure 14, with the spindle speed increases from 2500 rpm to 4000 rpm, the surface roughness (Ra) of sidewall decreases from 1.14 μm to 0.92 μm.

.

Question 10:

Results do not verify the 40 degrees as there is no theoretical work or simulation showing this, please support your statement – line 350

Answer 10:

Thanks for your professional comment. We have rewritten the sentences:

Experiments verified that a pulsating electrolyte generated by the wedged angle of 40° was more suitable for EC-milling process, and both the machining accuracy and surface quality were improved.

Question 11:

Please explain: in the simulation 40 degrees tool shows less turbulence than larger values; why lower turbulence, i.e. less effective removal of heat and by-products, generates a better result?

Answer 11:

Thanks for your professional comment. As you mentioned, in the simulation, it can be observed that the pulsating amplitude of electrolyte increases but the minimum velocity of electrolyte decreases even to about 0 m/s with the increase of wedged angle. Thus, anodic dissolution occurred in an instant static flow, which is not beneficial to the machining process. To make it clear, we have rewritten the sentences:

Combined with flow field simulation result, it can be explained that the pulsating amplitude of electrolyte increases but the minimum velocity of electrolyte decreases even to about 0 m/s with the increase of wedged angle. There is a low velocity zone of electrolyte in the IEG. Thus, anodic dissolution occurred in an instant static flow, the electrolytic by-products cannot be removed out of IEG, which is not beneficial to the machining process. In fact, many studies have reported that a pulsating flow with proper pulsating parameters is helpful to the transfer process [22].

Reviewer 3 Report

Authors have done interesting and useful work to fabricate deep-narrow grooves. The manuscript has merit and can be accepted after minor revisions. The suggestions/comments are as follows:

·         Wedge angle of the tube can affect the electrolyte pulsating flow field parameters. But how authors have chosen only 40°, 50°, and 60° wedge angles? Why not in the interval of 5° or why not they have chosen in the range of 0° to 90° in simulations and can be validated at 2-3 wedge angles experimentally?

·          a proper pulsating electrolyte can be generated in the IEG during the machining process by using 262 a wedged end face tube electrode with wedged angle of 40°”. The reason is not technical. Kindly support this argument with physics or mechanism. What does it mean by proper pulsating electrolyte?

·         In Fig. 11, indicate the taper angle and width for better representation/clarity.

·         Heading of subsection 4.4 is not correct. What does it mean by fabrication of EC milling? Fabrication of curved grooves/linear groove using EC milling….. or something similar. Here authors are fabricating grooves, not EC milling.

Overall manuscript has merit and can be accepted after minor revisions.

Author Response

Question 1:

Wedged angle of the tube can affect the electrolyte pulsating flow field parameters. But how authors have chosen only 40°, 50°, and 60° wedged angles? Why not in the interval of 5° or why not they have chosen in the range of 0° to 90° in simulation and can be validated at 2-3 wedged angles experimentally?

Answer 1:

In the preliminary experiment, less wedged angle has no significant effect on the experimental results. And there is larger low velocity zone of electrolyte and uneven electric field by using larger wedged angle, leading to the deterioration of machining. Thus, we have chosen 40°, 50° and 60° wedged angles in the simulation and experiment.

Question 2:

“a proper pulsating electrolyte can be generated in the IEG during the machining process by using a wedged end face tube electrode with wedged angle of 40°”. The reason is not technical. Kindly support this argument with physics or mechanism. What does it mean by proper pulsating electrolyte?

Answer 2

Thank you for your professional comment. We have rewritten the sentences:

The reason can be explained that: a pulsating electrolyte is generated in the IEG by using the wedged end face tube tool, which occurs flow separation resulting in a larger number of vortices at the wall surface and increasing the turbulence and mixing of fluid. The boundaries of this are characterized by the creation and destruction of eddies of large turbulence energy and vortex shedding, which helps the removal of electrolytic by-products and reduces the difference of electrolyte conductivity in the IEG along the depth direction of DNG [19].

Question 3:

In Fig. 11, indicate the taper angle and width for better representation/clarity.

Answer 3:

Thanks for your comment. Figure 11 has been indicated the SEM of sidewall of grooves. Thus, it is unable to indicate the taper angle and width in Fig. 11. Figures 9 and 10 have been show the taper and width of grooves.

Question 4:

Heading of subsection 4.4 is not correct. What does it mean by fabrication of EC milling? Fabrication of curved grooves/linear groove using EC milling….. or something similar. Here authors are fabricating grooves, not EC milling.

Answer 4:

Thank you for your comment. We have revised the heading of subsection 4.4:

EC-milling of complex narrow grooves by using wedged tube electrode